# Countering Vaccine Hesitancy among Pregnant Women in England: The Case of Boostrix-IPV

**DOI:** 10.3390/ijerph17144984

**Published:** 2020-07-10

**Authors:** Mairead Ryan, Laura A V Marlow, Alice Forster

**Affiliations:** Department of Behavioural Science and Health, University College London, London WC1E 6BT, UK; mairead.ryan.16@ucl.ac.uk (M.R.); l.marlow@kcl.ac.uk (L.A.V.M.)

**Keywords:** whooping cough, message framing, theory of planned behaviour, vaccine hesitancy

## Abstract

This study explored the effects of message framing on vaccine hesitancy for the antenatal whooping cough vaccine. The study also assessed whether the Theory of Planned Behaviour (TPB) constructs had any explanatory utility for vaccine intentions and behaviours in pregnant women. A between-subjects, cross-sectional design was employed. Participants (*n* = 282) were women who were pregnant (mean = 28 weeks, SD = 7.0), living in England and between 18 and 44 years of age. A self-report web-based survey was used to collect data. Participants were randomly assigned to read either (i) disease risk, (ii) myth busting, or (iii) control information before answering questions based on the TPB. No significant effects of message framing were found. Attitudes (Beta = 0.699; *p* < 0.001) and subjective norms (Beta = 0.262, *p* < 0.001) significantly predicted intention to vaccinate but perceived behavioural control did not. The TPB constructs accounted for 86% and 36% of the variance in vaccine intention and vaccine history respectively. Disease risk information did not influence vaccine acceptability in this sample of English pregnant women. The study offered preliminary evidence that interventions targeting constructs from the TPB may promote vaccine acceptability among pregnant women.

## 1. Introduction

Pertussis (whooping cough) is one of the leading causes of vaccine-preventable deaths worldwide [1]. Infants less than two months of age are the most vulnerable age group for pertussis-related morbidity and mortality [2,3]. In 2012, a national outbreak of whooping cough was declared in the UK. Over 9300 cases were reported, and 14 infants less than 12 months old died [4], resulting in the highest pertussis-related mortality rate since 1982 in the UK. In response to this epidemic, a pertussis-containing vaccine for pregnant women was introduced in October 2012 and was offered at routine antenatal appointments. The vaccine includes low dose diphtheria, tetanus, acellular pertussis, and inactivated poliomyelitis (Boostrix-IPV or DTaP/IPV). This vaccine is now routinely recommended in several countries (e.g., the US, Brazil, Israel, Belgium) and is available free of charge for pregnant women in the UK from 16-weeks gestation [5]. Despite the strong body of evidence providing support for the efficacy and safety of Boostrix-IPV, immunisation coverage, defined as the proportion of eligible persons who receive the vaccine, is suboptimal. Although immunisation coverage has increased since the introduction of the vaccine [6], it remains below the World Health Organisation coverage target of 90% worldwide [7], and the most recent annual report indicates that coverage in England is suboptimal and declining (69%) [8].

Vaccine hesitancy, defined as the ‘delay in acceptance or refusal of vaccination despite availability of vaccination services’ [9], is a central public health concern and pregnant women have been found to be more vaccine hesitant than the general population [10]. A systematic review addressing vaccine attitudes during pregnancy [11] reported that the majority of related studies have focused on the influenza vaccine in North America and the tetanus vaccine in Asia and Africa. The main concerns participants cited across all antenatal vaccines related to the safety and necessity of vaccines during pregnancy and low knowledge of the disease or vaccine. Only seven of 155 studies examined the whooping cough vaccine. Of those that did, most studies employed quantitative cross-sectional designs and examined a broad range of outcomes, from healthcare provider recommendations in the US [12] to immunisation coverage in Belgium [13] to vaccine intentions in Mexico [14].

Subsequent qualitative research which investigated attitudes toward Boostrix-IPV in the UK [15,16] reported similar findings to those cited across all antenatal vaccines in the review. The greatest self-reported barriers to uptake were safety concerns for the baby [16] and uncertainties relating to the risk and benefit of the whooping cough vaccine [15]. Despite studies which have identified consistent sources of vaccine hesitancy among pregnant women, there is a paucity of research which has developed and evaluated interventions to counter antenatal vaccine hesitancy for the whooping vaccine.

While it could be expected that any information to support the safety and benefit of vaccines would promote positive vaccine attitudes and increase immunisation coverage, research has highlighted that not all pro-vaccine messages are effective [17]. Nyhan and colleagues [17], for example, found that vaccine messages which directly addressed vaccine misconceptions and corrected vaccine myths reduced participants’ intentions to vaccinate. Other attempts to dispel vaccine safety misconceptions have also been ineffective and only served as a reminder of incorrect information to respondents [18].

The manner in which vaccine information is framed has previously been shown to significantly impact vaccine acceptability and uptake [19]. Many health researchers have drawn on Prospect Theory [20] to account for such findings, which maintains that presenting logically equivalent information about risk in different ways alters people’s preferences and actions. Several studies which employed loss-framed vaccine messages, highlighting the risk of not engaging in a protective behaviour [20], reported significant findings for increasing vaccine intentions over other strategies [21,22,23]. Horne and colleagues [23], for example, noted the success of messages which highlighted the consequences of failing to vaccinate. Hence, rather than attempting to refute common myths associated with the measles, mumps, and rubella vaccine, an approach which has previously been shown to be ineffective, they emphasised the risks of not vaccinating. They found that presenting this ‘disease risk information’ was significantly more effective in increasing vaccine acceptability (attitudes and intentions) than ‘myth busting information’; educational messages dispelling vaccine myths.

Additional theoretical frameworks which have been applied to understand and predict vaccine intentions and behaviours are constructs from the Theory of Planned Behaviour (TPB) [24]. The TPB posits that attitudes, subjective norms, and perceived behavioural control predict an individual’s intention to engage in a behaviour, which in turn, is the single best predictor of performing a behaviour [25,26]. Attitudes are defined as the degree to which a person has a favourable or unfavourable evaluation of an action. Subjective norms relate to normative influences and the perceived social pressure to perform or not perform an act. Perceived behavioural control (PBC) is the perceived ease or difficulty of performing the behaviour of interest, and PBC has been found to affect behaviour both directly and indirectly, through its effects on intentions [27]. The TPB has been criticised for its ability to better predict intentions over behaviour [28], its unsubstantiated assumption about the causal relationship between intention and behaviour [29], and disregard for other likely unconscious determinants of behaviour, such as emotions [30]. Nevertheless, the TPB has outperformed other theories in accounting for variance in vaccine uptake [31] and has been identified as a suitable theoretical model for predicting uptake for several vaccines [32,33].

The present study adopted a similar experimental design to Horne and colleagues [23] to explore the impact of framed messages on vaccine acceptability for Boostrix-IPV among a sample of pregnant women. The primary aim of the study was to assess whether disease risk information resulted in greater vaccine acceptability than myth busting information. Vaccine acceptability was measured using constructs from the TPB; attitudes, subjective norms, PBC and intentions. The secondary aim of the study was to assess whether constructs from the TPB were significantly associated with antenatal vaccine intentions and behaviours for Boostrix-IPV. Finally, the association between framing effects and background factors including, education level, number of children and vaccine history, was evaluated.

## 2. Materials and Methods 

A cross-sectional between-subjects design was employed whereby participants were randomly assigned to one of three experimental groups. Participation was voluntary, but respondents were offered an incentive of entering a draw to win a £20 online Amazon.co.uk voucher. A self-report web-based questionnaire via SurveyMonkey.com was used to collect data. Two attention checks were included in the survey to ensure thoughtful answers were provided. 

All participants were recruited via online social media platforms, namely Facebook.com and Netmums.co.uk. Participants had to identify as female, be at least 16 weeks pregnant, live in England and be between 18 and 50 years of age. The rationale for solely including participants who were 16 weeks pregnant was to ensure that all participants were eligible for the vaccine. A combination of targeted paid advertising and free informal advertising within online pregnancy groups was used to recruit participants. A total sample size of 279 was required by G* Power (Version 3.1. 9.2), University of Kiel, Germany, 2014 [34] to detect a small to medium effect size (f =0.1876), assuming a power of 0.80 and a two-sided significance of 0.05.

Ethical approval was granted by University College London Research Ethics Committee (reference: 10353/001). All data were collected between May and July 2017 inclusive. Respondents were briefed about the nature of the research via an information sheet before they were invited to participate in the study. After consent was obtained, participants were randomly assigned to one of three experimental conditions: (a) disease risk, (b) myth busting or (c) control. In each of the conditions, participants were asked to read brief information (less than one A4 page). In the disease risk condition, participants read a short story about a baby who contracted whooping cough, which highlighted the health risks posed by failing to vaccinate. In the myth busting condition, participants read that some pregnant women have safety concerns about the vaccine. They were then presented with a summary of three studies which refuted the link between vaccinating while pregnant and adverse outcomes for the mother and baby. In the control condition, participants read a vignette on an unrelated scientific topic (bird feeding). The content for all conditions was adapted from Horne and colleagues’ study [23]. Prior to data collection, the reading ability required for each condition was examined using three validated and widely used measures—the SMOG [35], the Gunning Fog [36] and the Flesch–Kincaid reading ease score [37]. This was to ensure the language was accessible and consistent across experimental conditions (see (Appendix A) for the wording of each condition).

After reading this information, participants’ vaccination attitudes were assessed based on constructs from the TPB [24]—PBC, attitudes, subjective norms, and intentions. Demographic information including age, ethnicity, education level, employment status, relationship status, gestational age, and number of children was also collected. Finally, participants were asked if they were aware of the vaccine prior to the study and whether they had already received Boostrix-IPV during their pregnancy (see Appendix B). To account for any potentially harmful effects of the experimental conditions, all participants were informed about the importance of the antenatal whooping cough vaccine in the debriefing sheet and directed to the National Health Service website for further information about the vaccine.

All analyses were conducted using SPSS Statistics for Windows, Version 24; IBM Corp Armonk, NY: 2016. The online survey was designed so that almost all questions demanded a response from participants. Data were missing for 4% of participants for one variable; missing cases were excluded from analyses involving that variable. 

Negatively phrased items were reversed, and summed scores were created for each TPB component, with higher scores indicating more positive attitudes, subjective norms, PBC and intentions towards vaccination. Cronbach’s alpha for each scale were acceptable (attitudes: α = 0.93; subjective norms: α = 0.82; PBC: α = 0.71; intentions: α = 0.98). The data violated assumptions of normality, even if transformed, so Kruskal–Wallis non-parametric analysis of variance was used to explore TBC scores across the three experimental groups. Data were also stratified to test for between-group differences, including education level (degree vs. no degree), previous children (yes vs. no), and already received whooping cough vaccine (yes vs. no). 

Among unvaccinated participants, multiple regression analyses were used to determine whether attitudes, PBC and subjective norms predicted intentions to receive the vaccine. Finally, logistic regressions were conducted to assess whether the four TPB constructs; PBC, subjective norms, attitudes, and intentions were significant predictors of vaccine history across all participants. The *p*-value for all analyses was set at a significance level of <0.05.

## 3. Results

### 3.1. Sample Characteristics

In total, 414 participants responded to the study invitation and 292 met all eligibility criteria. Participants who failed the attention checks (*n* = 10) were excluded from further analyses, resulting in a final sample size of 282 participants. Ninety-seven participants were randomly assigned to the disease risk condition, 98 to the myth busting condition and 87 to the control condition. The participants’ mean age was 31 years (SD = 5.13, range: 18–44 years). The majority self-identified as White British (82%), were educated to a university degree level or higher (72%), were employed (80%), married (67%), and had no children (63%). The participants’ mean gestational age was 28 weeks (SD = 7.08, range: 16–42 weeks). Almost all participants were aware of the whooping cough vaccine prior to the study (97%) but a smaller proportion had received the vaccine (55%). Please see Table 1 for all sample characteristics. Most participants reported positive attitudes and PBC to vaccination. Participants were more likely to report that they are influenced (a great deal/ a lot) by what their doctor thinks they should do (56%), as opposed to their parents (14%) or their female friends (13%). Summary descriptives for the TPB variables are presented in Table 2.

### 3.2. Main Effects of Message Framing

It was hypothesised that individuals in the disease risk condition would report significantly higher scores for each of the TPB scales than participants in the myth busting or control conditions, indicating more positive attitudes, subjective norms, PBC, and intentions. A series of Kruskal–Wallis H tests were conducted to test this hypothesis. Vaccinated participants’ intentions scores were excluded from this analysis. No statistically significant differences were found across the three conditions in attitudes (χ2 (2) = 3.57, *p* = 0.17), PBC (χ2 (2) = 0.11, *p* = 0.95), subjective norms (χ2 (2) = 0.36, *p* = 0.83), or intentions (χ2 (2) = 1.16, *p* = 0.56; Table 3). Data were stratified to test for between-group differences, including education level (degree vs. no degree), previous children (yes vs. no), and already vaccinated (yes vs. no). All between group differences remained non-significant. Mean intention scores were very high across all three conditions, with more than 80% of participants reporting that they were ‘very likely’ to receive the vaccine—Agree strongly/Agree moderately/Agree a little.

### 3.3. Determinants of Intention to Receive the Whooping Cough Vaccine 

A multiple regression was calculated to determine the influence of PBC, attitudes and subjective norms on intention to receive the whooping cough vaccine among participants who reported they had not already received the vaccine (*n* = 123). A significant regression equation was found; F (3119) = 235.98, *p* < 0.001, with an R2 of 0.856. Attitudes (Beta = 0.699, *p* < 0.001) and subjective norms (Beta = 0.262, *p* < 0.001) significantly predicted intention to vaccinate but PBC did not (Beta = 0.014, *p* > 0.05).

### 3.4. Determinants of Being Vaccinated against Whooping Cough

A multivariate logistic regression was conducted to assess whether the TPB constructs were significant predictors of having been vaccinated against whooping cough (*n* = 159) versus not being vaccinated (*n* = 123). The model which included PBC, attitudes, intentions and subjective norms was significant (χ2 (4) = 87.45, *p* < 0.001) and a good fit for the data (Hosmer and Lemeshow test; χ2 (8) = 4.96, *p* = 0.76). It correctly classified 72% of cases and explained approximately 36% of the variance (Nagelkerke R2 = 0.357). Both intention (odds ratio (OR) = 1.67, confidence interval (CI) = 1.17–2.39) and PBC (OR = 1.25, CI = 1.06–1.47) were significantly associated with being vaccinated. However, in contrast to determinants of intention to receive the vaccine, attitudes (OR = 0.93, CI = 0.80–1.08) and subjective norms (OR = 0.96, CI = 0.90–1.04) were not predictive of being vaccinated against whooping cough. 

## 4. Discussion

This study assessed the effects of message framing on vaccine hesitancy for the antenatal whooping cough vaccine. The primary aim of the study was to assess whether disease risk information better predicted vaccine acceptability than ‘myth busting’ information. The study also explored whether the TPB constructs had any explanatory utility for vaccine intentions and behaviour among a population of pregnant women. 

Contrary to this study’s hypothesis, and previous findings [21,22,23], no significant effects of message framing on vaccine acceptability were found across the three experimental conditions. These findings may be explained by factors unique to the current study. Previous studies have reported found that framing effects may depend on pre-existing characteristics of the participants [19,38]. Almost all participants were aware of the vaccine prior to the study. Given that all respondents were at least 16 weeks pregnant, this may be as a result of a recent health care consultation or otherwise. Health care consultations have previously been found to have a significant effect on vaccine acceptability [11,39,40,41,42]. Thus, it is possible that brief framing messages, such as those examined in the current study, are ineffective in altering vaccine attitudes among informed message recipients. These findings raise important questions for future research to address, given that the majority of message framing studies have not assessed participants’ prior vaccine-related knowledge [19]. 

Second, the effects of framing may be moderated by whether the advocated health behaviour is vaccination of the message recipient (i.e., mother) or vaccination of an individual for whom the recipient is responsible (i.e., baby/child) [43]. The antenatal whooping cough vaccine presents a unique case, in which the message recipient is both receiving the vaccine and, by proxy, choosing for someone else to be vaccinated. The effects of loss-framed messages have also been found to differ across vaccine type [19].

Results remained non-significant after adjusting for education level (degree vs. no degree), offspring status (children vs. no children) and vaccine history (yes vs. no). These results were consistent with previous research, which found no effect of offspring status [21] or vaccine history [44] on the efficacy of loss-framed vaccine messages. There is a dearth of research available with which to compare the findings of this study concerning the effect of education level on the efficacy of framed vaccine messages. However, a systematic review reported that the effects of education level on vaccine uptake, attitudes, and knowledge differed across and within countries, and thus, the authors concluded that education level cannot solely determine vaccine attitudes and behaviours [45].

The secondary aim of the study was to assess whether the TPB had any explanatory utility for vaccine intention and behaviour among pregnant women. The TPB was a good fit for the data and accounted for a significant proportion of the variance in both intention and behaviour. The TPB constructs predicted 86% of the variance in vaccine intentions. Subjective norms and attitudes emerged as significant predictors of vaccine intention, but PBC did not. Participants with more positive vaccine attitudes and those who perceived higher levels of approval from close family and friends showed stronger intentions to receive the antenatal vaccine. Consistent with previous meta-analyses [27,46,47] and other TPB vaccine research [32,33], the results suggest that the TPB presents an appropriate theoretical framework for interventions aimed at increasing antenatal vaccine intentions via specific proximal psychological determinants; subjective norms and attitudes. 

The four constructs of the TPB explained 36% of the overall variance in vaccine behaviour, with intention and PBC emerging as significant predictors of behaviour. This figure compares favourably with previous meta-analyses, which found that on average the TPB accounted for 27% of the variance in behaviour [27] and 19.3% of the variance in health-related behaviours [47]. These findings are encouraging; however, it must be highlighted that the TPB was better at predicting vaccine intentions than behaviour, the latter of which is of more interest for designing public health interventions. Furthermore, a considerable proportion of vaccine behaviour remains unaccounted for using the TPB constructs. Thus, although the current study offered preliminary evidence that the TPB constructs may be targeted to increase antenatal vaccine uptake, further research is needed to identify other contributing variables [28]. For example, vaccines are often considered an emotional decision for pregnant women [48], which the TPB does not account for.

Almost all of the participants (97%) were aware of the whooping cough vaccine prior to the study. The National Health Service has made this cohort a priority for public health action [49] and the responses in this study may reflect such public health campaign efforts. Nevertheless, awareness of the vaccine was markedly higher in this sample than relatively recent English study samples (e.g., 51.3%) [15], which were of similar biological and gestational age. The proportion of highly informed, educated, White British and married women was overrepresented in the current sample, reflecting a narrow range of demographics and hence, the responses may not be representative of the pregnant population in England. Previous research examining uptake for the MMR vaccine in the UK has found that parents who continue to reject vaccination for their child are from more advantaged backgrounds [50]. Thus, given that the sample in this study included pre-dominantly women from socio-economically advantaged backgrounds, who may be less susceptible to the effects of message framing, the efficacy of message framing on vaccine hesitancy should be explored in other populations. 

Mean intention scores were very high across all three conditions, with more than 80% of participants reporting that they were ‘very likely’ to receive the vaccine. A considerably smaller proportion of participants reported having already received the vaccine (55%). Pregnant women may receive the vaccine from 16 weeks up until the time of delivery, but they are encouraged to do so between 20 and 32 weeks, usually after the foetal anomaly scan. Of those who had not yet received the vaccine (123/282 participants), more than half were at or had surpassed the 32-week mark (56%), thereby raising uncertainty of whether they will receive the vaccine.

This study was limited by several noteworthy factors which should be addressed in future research to test the robustness of the findings. First, although participants from the relevant target population were recruited, where the majority of studies in this field have recruited university students [19], the final sample represented a narrow range of demographics, limiting the generalisability of the findings. Participants were active online pregnancy support group members and other social media users who volunteered to partake in vaccine research after being offered only a small incentive to do so. The number of women who elected not to participate in the study could not be measured, and hence, both the response rate and differences between respondents and non-respondents could not be determined. Future studies may benefit from recruiting participants from a diverse range of online platforms or settings. 

Second, this study was limited by the use of self-report measures rather than medical reports confirming vaccine receipt. The cross-sectional design of the study, used to reduce attrition, limited the ability of the study to determine the longer-term impact of framing messages. Participants who expressed intentions to receive the vaccine were not followed up to confirm receipt of vaccine. Cross-sectional studies are also not wholly appropriate designs to test the utility of the TPB as a model [51]. In the current study the TPB constructs were assessed for their ability to account for past rather than future behaviour. Finally, although an adequate overall sample size was achieved as per a priori power calculations, some participants had to be removed from the analyses because they had already been vaccinated. Hence, some of the results were underpowered. Whilst all participants were eligible at the time of participation to receive the vaccine (i.e., they were all at least 16 weeks pregnant), this resulted in many participants having already received the vaccine. Future researchers should determine participants’ vaccination status prior to recruitment to ensure that all analyses are sufficiently powered. 

## 5. Conclusions

Although significant framing effects were not observed in this cohort, the study made several important contributions to the field. To the authors’ knowledge, this was the first study to examine the effects of message framing on acceptability for the whooping cough vaccine among pregnant women. The study found no evidence that disease risk information affects antenatal vaccine acceptability. The results provided preliminary evidence that constructs from the TPB may be suitable targets for interventions aimed at reducing antenatal vaccine hesitancy.

Further research is needed to determine effective strategies to increase informed uptake in pregnant women. This cohort is an accessible population and is regularly exposed to health care information through routine antenatal care [52]. Pregnant women are also candidates for an increasing number of vaccines [52]. Hence, effective strategies for increasing uptake of Boostrix-IPV may provide useful insights for other antenatal vaccines.

## Figures and Tables

**Table 1 ijerph-17-04984-t001:** Sample characteristics.

	Disease Risk (*n = 97*)	Myth Busting (*n = 98*)	Control Condition (*n = 87*)
Age (years)			
18–24	14	6	11
25–34	63	65	51
35–44	20	27	25
Ethnicity			
White British	80	82	70
Other White	13	10	10
Non White	4	6	7
Education level			
GCSE or below	7	6	6
A Level or equivalent	19	9	6
College qualification	5	5	11
Degree or higher	66	78	64
Employment status			
Employed	74	84	71
Unemployed	23	14	16
Relationship status			
Single	5	3	4
In a partnership	32	26	18
Married	60	69	65
Previous children			
No	53	65	60
Yes	44	33	27
Gestational age (wks)			
16–24	29	40	27
25–34	41	32	39
35+	23	22	19
Vaccine awareness			
Yes	94	96	84
No	3	2	3
Received vaccine			
Yes	59	55	45
No	38	43	42

**Table 2 ijerph-17-04984-t002:** Summary descriptives for each Theory of Planned Behaviour (TPB) variable.

Items	Overall	Currently Vaccinated	Currently Unvaccinated
	*n* (%)	*n* (%)	*n* (%)
Perceived behavioural control ^a^			
If I wanted to, I could attend an appointment to get vaccinated against whooping cough.	264 (93.6)	158 (99.4)	106 (86.2)
I feel confident in my ability to get vaccinated for whooping cough.	259 (91.8)	159 (100)	100 (81.3)
There are (no) barriers in the way of me receiving the whooping cough vaccine.	235 (83.3)	142 (89.3)	93 (75.6)
Attitudes ^a^			
Getting vaccinated for whooping cough will help protect my baby from whooping cough.	245 (86.9)	157 (98.7)	88 (71.5)
If I get vaccinated for whooping cough, I can reduce my baby’s risk of whooping cough.	249 (88.3)	158 (99.4)	91 (74.0)
Getting vaccinated for whooping cough will decrease my chances of getting whooping cough.	216 (76.6)	137 (86.2)	79 (64.2)
Subjective norms ^b^			
How much do your parents think you should receive the whooping cough vaccine?	156 (55.3)	100 (62.9)	56 (45.5)
How much does your doctor think you should receive the whooping cough vaccine?	232 (82.3)	146 (91.8)	86 (69.9)
How much does your best female friend think you should receive the whooping cough vaccine?	173 (61.3)	109 (68.6)	64 (52.0)
In general, I want to do what my parents think I should do.	39 (13.8)	25 (15.7)	14 (11.4)
In general, I want to do what my doctor thinks I should do.	158 (56.0)	106 (66.7)	52 (42.3)
In general, I want to do what my best female friend thinks I should do.	37 (13.1)	22 (13.8)	15 (12.2)

^a^ Agree strongly/ Agree moderately/ Agree a little; ^b^ A great deal/A lot.

**Table 3 ijerph-17-04984-t003:** Mean Scores for the Theory of Planned Behaviour variables across the three intervention conditions (*n* = 282).

		Disease Risk (*n* = 97)		Myth Busting (*n* = 98)		Control (*n* = 87)		Kruskal-Wallis
	Range	Mean	SD	Mean	SD	Mean	SD	*p*-value
Attitudes	3–21	18.41	5.01	18.36	4.72	17.55	5.36	0.17
Subjective norms	6–30	14.13	5.27	13.98	4.89	13.64	5.26	0.83
Perceived Behavioural Control	3–21	19.23	3.18	19.08	3.38	19.39	2.85	0.95
Intentions	4–28	20.42	10.67	22.14	9.66	20.24	10.67	0.56

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
