# Peer review of "Countering Vaccine Hesitancy among Pregnant Women in England: The Case of Boostrix-IPV"

_ijerph, 2020, doi:10.3390/ijerph17144984_

Round 1

Reviewer 1 Report

    I recommend making a general review of the work including a more concrete and less diffuse development, because I think it is an interesting and valid topic, but more previous documentation is missing.

Reviewer 2 Report

This is a well-written and well-organized article about educational/behavioral interventions to address vaccine hesitancy among pregnant women receiving the pertussis vaccination. While the study did not show that disease risk information influenced vaccine acceptability, I do think the article has the potential to make an important contribution to the literature. However, there are some important study limitations that should be addressed or better explained prior to publication.

Introduction:

  1. I would expand on the first paragraph to give more background about the importance of the pertussis vaccination in pregnancy. When was a pertussis vaccine first introduced for pregnant women? Is this done routinely on a global scale? When is this typically done during the pregnancy? I know you answer this last question later in your discussion, but it seems to me that this is critical to your study. Since you enrolled women throughout the pregnancy, many women had already received their pertussis vaccination (over half). So why would your intervention have impacted their attitudes or decisions about the vaccination since they weren’t vaccine hesitant? This was very confusing to me and should be better explained.

Methods:

  1. Line 99: Ethical approval was granted by…
  2. Data should be plural (lines 100 and 164).
  3. Explain the interventions more. I know that they are included in the appendix, but most people will not see the appendices and you should briefly describe the interventions in the Methods section. For example, were they brief online information sheets? How long was each intervention? Was this information created by the study team? What was it based on (e.g., NHS information)? Why did you choose bird feeding and not something to do with pregnancy (but not related to vaccination)? Bird feeding seems very random.
  4. Lines 116-119: Was this debriefing sheet provided to everyone at the conclusion of the data collection? If not, it obviously would/could have affected your interventions. This needs to be explained in a clearer way.

Results:

  1. I am confused about why you included women who had already received the pertussis vaccination in your study. Over half of your participants were known to already be vaccinated. Plus, many of those who weren’t vaccinated were not yet at the typical gestational age to receive the vaccination. It seems like a better design would have been to enroll everyone early in the pregnancy, before they would have been vaccinated or to enroll those who weren’t vaccinated yet at a certain gestational age (e.g., lines 260-262). Can you do the analysis with just those in lines 260-62? A better explanation of your study design might help clarify this point as well.
  2. Table 1 should have p-values to determine any significant differences between groups.
  3. Table 2 should have p-values to determine any significant differences between groups.

Discussion:

  1. Were you adequately powered to determine a significant difference between three different study cohorts in a small pilot study with 80% of participants very likely to receive the vaccine (and over 50% already having received it)? There weren’t many vaccine hesitant individuals to try and influence. It seems to me that your study design may be why you didn’t have a significant effect.

Reviewer 3 Report

This article reports a survey undertaken by English scholars regarding pregnant women’s responses to different types of vaccine messaging regarding maternal pertussis vaccination. Amongst a sample of 282 pregnant women, it found no significant difference between women who were offered one of three framings, the third of which was a control information piece on how to feed birds.

The authors are to be encouraged for reporting their non-findings. We need to know what doesn’t appear to work as much as what does. However, the authors also note that inbuilt limitations of their study, particularly the over-representation of highly educated white women, may have distorted the findings. They represent their work as contributing to the growing literature on the impact of vaccine messaging and in particular on pregnant women, where they claim their study is the first to do so. The researchers may also be interested in the work of Elizabeth Helen Hayles and a study she led in Vaccine in 2014 about the short-lived cocooning strategy in Australia.

The authors have included a well-researched literature review and they link their findings back to existing literature. A gap in this appears to lie in engaging more deeply with pregnant women’s receipt of influenza vaccination. Although they are different vaccines, in many settings they are the two key pregnancy vaccines funded and recommended for pregnant women. Teasing out more existing knowledge about framing studies with regard to influenza vax in pregnant women would be helpful, as it would orient our thinking about how pertussis vax might be different and in what ways, and what we might learn from the influenza vax experience.

More context on the British vaccination system would be welcome too in an international journal. The NHS’s rigorous promotional role is described, but it is also important to tell us the context and setting where vaccination occurs (eg GP clinic, midwife clinic at hospital) and that it is free of charge to consumer (and since when).

As a qualitative methodologist, I can’t comment on the details of the quantitative methods. What I can do is comment on whether the quantitative details of the study are accessible and understandable to those who do not conduct this type of research. The authors need to better explain what the TPB is, rather than just launch into its cons and then pros. The authors also use the PBC acronym without ever explaining what it is, and this is a real problem. I would have appreciated a more straightforward and non-technical explanation of why and how the the theory of planned behaviour was a useful predictor of vaccination history.

Reviewer 4 Report

Please rephrase lines 28-29. It would sound better if you started the sentence with although instead of using however in the middle of the sentence. In lines 32 and 247 the percentage should not be in the bracket with the reference. It should be placed in parenthesis before the reference. In lines 51-57 it would be beneficial to the reader to add more vaccine hesitancy studies from Europe for comparison. The authors have cited studies with ineffective interventions to promote antenatal vaccination. However. there are studies regarding interventions to promote antenatal vaccination that show promising results (such as the one from Psarris et al. from Greece). The references need the authors attention (for instance 46 and 47 are missing the access date and many have the year of publication highlighted while other do not).
